# Potential Implication of Azole Persistence in the Treatment Failure of Two Haematological Patients Infected with *Aspergillus fumigatus*

**DOI:** 10.3390/jof9080805

**Published:** 2023-07-30

**Authors:** Teresa Peláez-García de la Rasilla, Álvaro Mato-López, Clara E. Pablos-Puertas, Ana Julia González-Huerta, Alicia Gómez-López, Emilia Mellado, Jorge Amich

**Affiliations:** 1Microbiology Department, Central University Hospital of Asturias (HUCA), 33011 Oviedo, Asturias, Spain; 2Institute for Health Research in the Principality of Asturias, Instituto de Investigación Sanitaria del Principado de Asturias (ISPA), 33011 Oviedo, Asturias, Spain; 3Mycology Reference Laboratory (Laboratorio de Referencia e Investigación en Micología LRIM), National Centre for Microbiology, Instituto de Salud Carlos III (ISCIII), 28220 Majadahonda, Madrid, Spain; 4Hematology-Stem Cell Transplantation Unit, Hospital Universitario Central de Asturias (HUCA), 33011 Oviedo, Asturias, Spain; 5CIBER de Enfermedades Infecciosas (CIBERINFEC-CB21/13/00105), Instituto de Salud Carlos III, 28029 Madrid, Spain; 6Manchester Fungal Infection Group (MFIG), Division of Evolution, Infection, and Genomics, Faculty of Biology, Medicine and Health, University of Manchester, Manchester M139NT, UK

**Keywords:** invasive aspergillosis, *Aspergillus fumigatus*, treatment failure, azole persistence, poly-genomic infections

## Abstract

Invasive aspergillosis (IA) is a major cause of morbidity and mortality in patients receiving allogeneic haematopoieticcell transplantation. The deep immunosuppression and a variety of potential additional complications developed in these patients result in IA reaching mortality rates of around 50–60%. This mortality is even higher when the patients are infected with azole-resistant isolates, demonstrating that, despite the complexity of management, adequate azole treatment can have a beneficial effect. It is therefore paramount to understand the reasons why antifungal treatment of IA infections caused by azole-susceptible isolates is often unsuccessful. In this respect, there are already various factors known to be important for treatment efficacy, for instance the drug concentrations achieved in the blood, which are thus often monitored. We hypothesize that antifungal persistence may be another important factor to consider. In this study we present two case reports of haematological patients who developed proven IA and suffered treatment failure, despite having been infected with susceptible isolates, receiving correct antifungal treatment and reaching therapeutic levels of the azole. Microbiological analysis of the recovered infective isolates showed that the patients were infected with multiple strains, several of which were persisters to voriconazole and/or isavuconazole. Therefore, we propose that azole persistence may have contributed to therapeutic failure in these patients and that this phenomenon should be considered in future studies.

## 1. Introduction

*Aspergillus fumigatus* is an opportunistic fungal pathogen that can cause respiratory infections in individuals with imbalances in the immune system [1]. In nature, this fungus produces thousands of small spores (2–3 µm) that are constantly inhaled by humans [1]. Whilst in healthy individuals this has no consequences, in immunocompromised patients, *A. fumigatus* can cause life-threatening invasive aspergillosis (IA) [1,2], which has overall mortality rates in the range of 20–50% [3,4,5,6,7]. Due to immune deficiencies, caused by both the underlying disease and/or the applied immunosuppressive therapies, haematological patients are at high risk of developing severe infections, with IA being one of the most common and devastating [8]. Indeed, IA is the most common fungal infection and a leading cause of infection-related mortality in haematopoietic cell transplant recipients [9,10]. The incidence of IA varies depending on the underlying haematological disease [11], but it has a remarkable overall incidence of 4–12% [8]. Moreover, the global 100-day case fatality rate in haematological patients has recently been calculated to be 29%, which is still quite significant [8].

Azoles are the first-line therapy for both pre-emptive and evidence-based treatment of aspergillosis diseases [12]. Worryingly, in the last two decades, there has been a surge and spread in resistance to azoles [13] and it has been described that infection with azole-resistant isolates is associated with higher mortality [14,15]. This is a cause of concern that has prompted the establishment of susceptibility profiling in the mycology diagnostic setting. Indeed, standard protocols have been published by the European Committee on Antimicrobial Susceptibility Testing (EUCAST, [16]) and Clinical and Laboratory Standards Institute (CLSI, [17]) to regulate and homogenize the diagnosis of antifungal resistance worldwide. This susceptibility profiling often guides the determination of the appropriate antifungal therapy for individual cases.

Additionally, IA mortality is unacceptably high in patients infected with azole-susceptible isolates, even in cases with correct diagnosis and appropriate antifungal treatment. There are several, non-mutually exclusive, reasons that can account for mortality in these cases. A major reason for treatment failure is inadequate levels of azoles in blood, due to poor absorption or patient PK/PD variability [18]. Therefore, close monitoring of drug levels is strongly recommended in IA patients [19,20,21], although, even in patients with sufficient drug levels, it is crucial that the azole reaches the focus of fungal growth in tissues, which can be difficult due to necrosis or blood vessel damage [22,23], and penetrates the fungal mass [24,25].

Finally, antimicrobial tolerance and persistence have been implicated in treatment failure in bacterial infections [26,27,28] and fluconazole tolerance has already been suggested to correlate with treatment failure in candidemia [29,30].

We have recently demonstrated that some *A. fumigatus* isolates can display persistence to azole antifungals [31]. We found that certain strains can survive for extended periods, and even grow at slow rates, at supra minimum inhibitory concentrations (MIC) of voriconazole (VRC) and other azoles (see Appendix A for a visual example). In a *Galleria mellonella* model, VRC treatment seemed to inadequately eradicate persister isolates in some larvae, whilst it was consistently effective with non-persister isolates. Therefore, we speculated that, in some individuals, infection with persister isolates may reduce the efficacy of azole therapy and, thus, it might contribute to treatment failure in some cases.

In this study we identified two haematological patients who presented proven IA and experienced treatment failure, despite being infected with susceptible isolates and receiving appropriate therapy that resulted in blood drug levels reaching therapeutic range. We show that these patients were infected by multiple *A. fumigatus* strains, some of which displayed persistence to the azoles employed. Therefore, we propose that azole persistence may have contributed to treatment failure in these patients.

## 2. Materials and Methods

### 2.1. Aspergillus fumigatus Strains

In this study, a total of 26 *Aspergillus fumigatus* clinical isolates were analyzed. All *Aspergillus* isolates were cultured according to routine mycological procedures. For DNA extraction, conidia from each strain were cultured in glucose–yeast extract–peptone (GYEP) liquid medium (0.3% yeast extract, 1% peptone; Difco,) with 2% glucose (Sigma-Aldrich) for 24 h at 37 °C. After mechanical disruption of the mycelium by vortex-mixing with glass beads, genomic DNA of isolates was extracted using the standard phenol–chloroform–isoamyl method. Molecular identification was performed by PCR amplifying and sequencing ITS1-5.8S-ITS2 regions and a portion of β-tubulin gene *benA* (AFUA_1G10910) [32].

### 2.2. Clinical Antifungal Drugs Susceptibility Testing

Antifungal susceptibility testing was performed following the European Committee on Antimicrobial Susceptibility Testing (EUCAST) broth microdilution reference method 9.3.1 [16]. The antifungals used were the azoles itraconazole (IZ), voriconazole (VRC), posaconazole (POS) and isavuconazole (ISV) and the polyene amphotericin-B (AMB) (Sigma, St. Louis, MI, USA). The final concentrations tested ranged from 0.016 to 8 mg/L for the four azoles and from 0.032 to 16 mg/L for AMB. Minimal inhibitory concentrations (MICs) were visually read after 48 h of incubation at 37 °C in a humid atmosphere. MICs were performed at least twice for each isolate. Clinical breakpoints established by EUCAST [33] were used to classify the *A. fumigatus* strains as susceptible or resistant.

### 2.3. Determination of Persistence

To determine the persister phenotype of the isolates, we employed a standard operating procedure recently optimized in our laboratory. Briefly, 2 × 10^5^ conidia of each isolate were incubated in 200 µL of RPMI-1640 (Sigma, St. Louis, MI, USA) containing 8 mg/mL of VRC or ISV in 96-well plates for 72 h at 37 °C. After incubation, the plates were centrifuged at 3500 rpm for 5 min, with the brake disabled, the media were discarded and the conidia resuspended in 200 µL of NaCl (0.5%)–Tween20 (0.002%) solution. The full content of the well was spread on PDA (Oxoid, Cheshire, UK) plates and incubated for 24 h at 37 °C, followed by another 24 h at room temperature. The number of colony-forming units (CFUs, i.e., surviving conidia) were then counted. In each experiment, two control strains were included, ATCC46645 as a non-persister and PD-9 as a persister [31]. The experiment was performed three independent times, with three replicates on each occasion. Isolates were classified as persister when the number of CFUs was >0.015% (less than 99.85% killing) for VRC and >0.02% for ISV (less than 99.98% killing).

### 2.4. Cyp51A and Cyp51B Amplification, PCR Conditions and Sequencing

The full *cyp51A* coding sequence, its promoter and the *cyp51B* coding sequence were PCR-amplified and sequenced. PCR reaction mixtures contained 0.5 μM of each primer, 0.2 μL of dNTPs 10 mM (Roche, Minato, Tokyo), 5 μL of PCR 10x buffer, 2 mM of MgCl_2_, DMSO 5.2%, 2.5 U of AmpliTaq DNA polymerase (Applied Biosystems, Waltham, MA, USA) and 100–200 ng of DNA in a final volume of 50 μL. A DNA 1 kb molecular ladder (Promega, Singapore) was used for all electrophoresis analyses. Samples were amplified in a GeneAmp PCR System 9700 (Applied Biosystems, Waltham, MA, USA). The parameters used were 1 cycle of 5 min at 94 °C and then 35 cycles of 30 s at 94 °C, 45 s at 56 °C for *cyp51A* promoter and 58 °C for *cyp51A* gene and then 2 min at 72 °C, followed by a final cycle of 5 min at 72 °C. The amplified products were purified using IllustraExoProStar 1–step (GE Healthcare Life, Singapore) and both strands were sequenced with the Big-Dye terminator cycle sequencing kit (Applied Biosystems, Waltham, MA, USA) following the manufacturer’s instructions. All gene sequences were edited and assembled using the Lasergene software package (DNAStar Inc., Madison, WI, USA). Primers used to amplify and sequence *cyp51A* and its promoter and the *cyp51B* gene have been previously described [34].

### 2.5. Strains Genotyping

All strains included in this study were genotyped following the previously described typing method TRESPERG [35]. Four markers were used: (i) AFUA_2G05150, encoding an MP-2 antigenic galactomannan protein (MP2); (ii) AFUA_6G14090, encoding a hypothetical protein with a CFEM domain (CFEM); (iii) AFUA_3G08990, encoding a cell surface protein A (CSP) and (iv) AFUA_1G07140 (ERG4B), which encodes a putative C-24(28) sterol reductase. The combination of the genotypes obtained with each marker has a discriminatory value (D) of 0.9972.

### 2.6. Monitoring Antifungal Levels

The blood levels of ISV were checked retrospectively using a liquid chromatographic method coupled with UV detection adapted from a multiplex validated method [36].

## 3. Results

### 3.1. Patient 1

A 21-year-old man was diagnosed with acute promyelocytic leukemia. He was treated with several lines of chemotherapy and, finally, in the third complete remission, the patient underwent an allogeneic haematopoietic cell transplant (allo-HCT) from an unrelated donor, without complications.

On day +54 post-transplant, the patient developed acute graft versus host disease (aGVHD) grade III that required intense immunosuppressive treatment (high-dose corticosteroids, tacrolimus, mycophenolate, extracorporeal photopheresis and ruxolitinib). As this treatment increased the risk of infection [37,38], he maintained anti-infectious prophylaxis with acyclovir, POS (tablets, standard dose of 300 mg/day) and trimethoprim–sulfamethoxazole, until the immunosuppressive regimen ended. During admission, he presented with CMV gastrointestinal disease and BK hemorrhagic cystitis.

Ten months after allo-HCT, the patient presented a chronic GVHD leading to Bronchiolitis Obliterans Syndrome (BOS) and treatment with FAM therapy (Fluticasone, Azithromycin and Montelukast) was initiated.

Sixteen months after allo-HCT, the patient report a marked dyspnea, without fever or expectoration. Suspecting exacerbation of BOS, treatment was started with prednisone (0.5 mg/kg) and levofloxacin. The thoracic computed tomography (CT) image showed multiple cavitated nodules in both upper and lower lobes with ground glass density.

Treatment was started with VRC and liposomal amphotericin B (AMB). The BAL samples were positive for *Aspergillus* (PCR, GMN and LFD) with isolation of pan-susceptible *A. fumigatus* in the culture.

In the first 10 days, the patient presented good clinical evolution, remaining afebrile and with a decrease in blood galactomannan. However, on day +3, it was necessary to discontinue treatment with VRC due to visual hallucinations and it was decided to maintain AMB in combination with anidulafungin (AND). On day +13, AMB was discontinued, as the patient suffered a worsening chronic renal insufficiency; treatment with ISV was started while administration of AND was maintained (Figure 1A).

In the following 10 days (day +22), his condition markedly worsened, with recurrence of fever, cough and haemoptoic sputum, as well as increased GMN and PCR values again. This appeared to coincide with the discontinuation of AMB treatment and so the drug was restarted with close monitoring of renal function. Inhaled amphotericin B was also added and ISV maintained.

In the seven weeks following admission, the patient showed a slow but progressive clinical improvement, with decreasing control parameters (GMN and PCR) and stabilization of radiological images (Figure 1A). However, the patient then developed a subsequent hemorrhagic shock that required admission to the ICU with mechanical ventilation. After 10 days, since it was determined that extubation would not be possible, it was decided to limit therapeutic effort and the patient ultimately died on day +72. Histopathologic examination of several specimens obtained by biopsy, in which hyphae were seen accompanied by evidence of associated tissue damage, confirmed the aspergillosis diagnosis.

In this patient, a total of 14 *Aspergillus* isolates were recovered from 4 sputum (SP) samples, 1 tracheal aspirate (TA) and 2 bronchoalveolar lavages (BAL) taken throughout the treatment period (Table 1). Partial *benA* gene sequencing identified the isolates as *A. fumigatus senso stricto*. The blood levels of ISV were checked retrospectively and confirmed to be in the therapeutic range throughout treatment (>2 mg/L [39,40], Figure 1B and Appendix A). All *A. fumigatus* isolates were subjected to susceptibility testing, using the broth microdilution method according to the EUCAST E.Def. 9.3 instructions [16] and were typed using the TRESPERG methodology [35]. It was found that the patient was infected by at least five different isolates (TYPES I to V, Figure 1C and Table 1) and all of them (except the last isolate (TYPE V), which appeared very late in the course of infection (day +64)) were susceptible to all antifungals tested (Table 1). Sequence of the genes *cyp51A* and *cyp51B* revealed that the TYPE V isolate carries the TR34/L98H mutation in *cyp51A*, which explains its resistance profile (Table 1). We then checked if the isolates displayed antifungal persistence to VRC and ISV using our recently established protocol and found that the TYPES III and IV were persisters to both VCZ and ISV (Table 1 and Appendix A).

### 3.2. Patient 2

A 37-year-old woman was diagnosed with acute myeloblastic leukemia with myelodysplasia-related changes and adverse cytogenetic risk ELN 2017. She received intensive chemotherapy treatment followed by an allo-HCT from an unrelated donor with myeloablative conditioning, without remarkable complications.

After 27 days following the transplant, the patient developed aGVHD global GII with complete response to a short cycle of systemic corticosteroids at 2 mg/kg. Prophylaxis against filamentous fungus was maintained with POS (tablets, standard dose of 300 mg/day). She did not present any relevant infectious complications during that time.

Nine months after transplant, a diagnosis of BOS was proposed in connection with cGVHD, and prednisone (1 mg/kg) was initiated with MMF, FAM therapy and ruxolitinib; prophylaxis with oral POS was also restarted.

One year after allo-HCT, the evolution continued to be poor. A new line of immunosuppression with sirolimus was prescribed and, at the same time, antifungal prophylaxis was adjusted, replacing POS with ISV. However, in month +13 after transplant, the patient suffered a marked deterioration in pulmonary function that required hospitalization. A bronchoscopy with bronchoalveolar lavage (BAL) was carried out, in which infections by atypical bacteria, mycobacteria and *Pneumocystis jirovecii* were ruled out, but a positive LFD was obtained. A chest CT was performed; however, as no compatible lesion was detected, the ISV prophylaxis was not modified.

One month later, due to exacerbation of purulent respiratory secretions, a sputum sample was taken (day +1). As this showed growth of *A. fumigatus* and all fungal biomarkers (LFD, PCR and GMN) were positive (Figure 2A), treatment with ISV was supplemented. High-resolution computed tomography (HRCT) showed lesions compatible with invasive pulmonary aspergillosis. Therefore, this infection can be considered a case of breakthrough aspergillosis, as the patient was under ISV prophylaxis for around two months.

The sputum taken on day +6 showed new growth of numerous colonies of *A. fumigatus*, with positive LFD, PCR and GMN testing (Figure 2A). On day +12, after a week of combined treatment, a new sputum sample was collected with negative *Aspergillus* LFD, lower GMN and fewer copies of *Aspergillus* PCR compared to the previous sample. However, the clinical evolution continued to be poor, with increased need for O_2_ and respiratory work. GMN remained negative in blood samples throughout the evolution, although the patient was not neutropenic.

After two weeks, the patient exhibited respiratory failure, requiring admission to ICU with orotraqueal intubation, and nebulized AMB was added to the treatment. The status of the fungal infection was reevaluated with a new sample of TA (day +18), which was positive for culture and biomarkers (PCR and LF) of *A. fumigatus* (Figure 2B). HRCT was performed again, which showed multiple bilateral pulmonary nodules of thickened wall, a halo of density in “ground glass” that had increased significantly in number and size with respect to the previous images.

The progression in the ICU was negative, with mechanical ventilation becoming necessary. TA (day +29 and day +30) were positive for *A. fumigatus* culture and biomarkers. After seventeen days in the ICU, the patient’s condition gradually worsened and the respiratory system declined, ultimately causing exitus. Post-mortem histopathologic examination of the tissue confirmed the diagnosis of aspergillosis.

In this patient, a total of 12 *Aspergillus* isolates were recovered from 3 SP and 3 TA taken throughout the treatment period (Table 2). Partial *benA* gene sequencing identified the isolates as *A. fumigatus senso stricto*. The patient started antifungal prophylaxis 62 days before the IA diagnosis was made (Figure 2A). The blood levels of ISV were checked retrospectively and confirmed to be in therapeutic range throughout its administration (>2 mg/L [39,40], Figure 2B and Appendix A). Remarkably, the levels decreased abruptly around the time of diagnosis and remained lower (1–3 mg/L) during the infection course. (Figure 2B and Appendix A). All *A. fumigatus* isolates were subjected to susceptibility testing, using the broth microdilution method according to the EUCAST E.Def. 9.3 instructions [16] and were typed using the TRESPERG methodology [35]. It was found that the patient was infected by at least three different isolates (TYPES I to III) (Figure 2C and Table 2) and all three were susceptible to every antifungal tested. In agreement, sequencing of the genes *cyp51A* and *cyp51B* showed that all isolates carried wild-type alleles. Therefore, we checked whether the isolates could be persisters against VRC and ISV using our recently established protocol. Interestingly, we found that TYPE II and III, which were isolated only once, were persisters to both VRC and ISV (Table 2 and Appendix A). In addition, the TYPE I isolates displayed a variable phenotype but were detected as persisters to ISV for most independent isolates. Despite treatment with appropriate antifungals, the patient always presented high PCR and galactomannan (GMN) biomarker values (Figure 2A).

## 4. Discussion

In this study, we have presented two cases of haematological patients who developed invasive aspergillosis. Despite good management, correct diagnosis and, consequently, adequate treatment, which resulted in therapeutic drug levels in their blood, the patients did not respond to therapy and reached exitus.

Undoubtedly, one major factor that has accounted for treatment failure in these cases is the severe immunosuppressed status of the patients. Invasive aspergillosis after allogeneic transplantation of hematopoietic progenitors, further complicated by GVHD, has associated mortality rates of around 50–100% [15,41]. Remarkably, when the infection is caused by resistant isolates, the mortality noticeably increases [41], demonstrating that an efficient antifungal treatment is still valuable and can provide a measurable benefit in these patient groups.

Our detailed characterization of these cases has revealed that the patients were infected by several different isolates with different susceptibility profiles. Infections with multiple *A. fumigatus* isolates have been reported before [42,43,44], but the potential implications of such poly-genomic infections have not been studied in detail. We believe that infections with mixed strains may have a strong impact on the efficacy of the antifungal therapy. Therefore, in all long-term treatments with antifungal agents, especially with azoles, repeated sampling and regular susceptibility testing of isolated strains is essential, even though this is not always easy to perform [12]. This is crucial, as individual isolates may have different susceptibility profiles and it is very difficult to predict the development of persistence and/or resistance that might occur in several isolates. This is not only true in cases of aspergillomas but also in invasive aspergillosis that develops in immunosuppressed patients, who often need treatment for several weeks or months. Indeed, in the presented cases, we found that the patients were infected by various susceptible, persistent and (in patient 1) resistant isolates. In the first case presented, on day +64, a persister and a resistant isolate were detected in the same sample (Table 1) and, in case 2, the same genotype showed different persister profiles over time (Table 2).

In these cases, both patients were infected by various isolates and all but one (the last isolate in patient 1) were susceptible to all of the antifungals (Table 1 and Table 2), which would indicate that the treatment administered could have been effective.

We have recently described that some *A. fumigatus* isolates can display persistence to azoles, which we described as their capacity to survive and even grow at slow rates in the presence of supra-MIC concentrations of the antifungal [31]. Therefore, we hypothesized that azole persistence may have played a role in the treatment failure experienced by these patients.

In patient 1, we found that two of the four infecting susceptible isolates (TYPES III and IV) were indeed persisters to VRC and ISV. Interestingly, non-persister TYPES I and II seemed to be efficiently eradicated, as they were not recovered again shortly after being detected; in contrast the persister isolates seemed to withstand therapy, as they were recovered throughout two months of antifungal treatment. Consequently, we speculate that the antimycotic therapy may not have been able to eradicate the persister isolates, resulting in a maintained infection, until a resistant isolate appeared and, ultimately, caused treatment failure. This resistant isolate was likely newly acquired from the environment, as suggested by its TR34/L98H mutation in *cyp51A*, which is known to develop in agricultural settings [45,46].

In case 2, the patient was infected by at least three different persister isolates, two of which were detected only once during the course of infection, suggesting that despite being persister they may have been properly eradicated by treatment, or that the TYPE I isolate had increased fitness in the infected tissues. The TYPE I isolate displayed various morphotypes as well as variable persistence capacities, although it was detected as persister to ISV in the majority of tests (7/10, Table 2). This variability could be due to multiple factors; for instance, intra-patient evolution of the isolate or incapacity of the TRESPERG method to distinguish between two very closely related strains. At an early stage, before the diagnosis of IA, the patient showed an abrupt drop in ISV blood levels (maybe caused by her deterioration requiring admission in ICU [47]), which we speculate may have allowed the establishment of infection by the TYPE I isolate (which occurred during ISV prophylaxis) and prevented its eradication, eventually causing treatment failure.

Remarkably, both patients received combinatorial therapy, as recommended in cases of bilateral aspergillosis and respiratory insufficiency [48] (additionally, patient 2 developed infection under ISV prophylaxis). Nevertheless, despite the expected additive effect of these antifungals, the persister isolates could not be eradicated by the treatment. One possible explanation would be if the strains showed persistence to AMB as well, something worthy of future study. In addition, we previously found that combinatorial treatment does not seem to prevent azole persistence [31], implying that sub-optimal levels of the combinatorial drugs in patients would not aid in the eradication of the isolates.

There are additional factors that could account for treatment failure in these patients, although they are not mutually exclusive with a potential role of persistence. For example, sufficient drug penetration at the site of infection is a key factor for the efficacy of antimicrobials [49,50], and it has been proposed that poor penetration can account for treatment failure in aspergillosis infections [51]. We cannot rule out that the drugs did not sufficiently penetrate the *Aspergillus* infection foci but, given that suitable ISV blood concentrations were achieved (Figure 1B and Figure 2B) and that ISV has been shown to have very good tissue distribution and penetration at the site of infection [24,52], we believe it is unlikely that poor penetration could be the only factor responsible for the inability to eradicate the persister isolates in these patients. In fact, relatively poor drug penetration, causing lower than expected drug levels at the foci of infection, would be synergistic with *A. fumigatus* azole persistence to trigger survival of the isolates.

Another factor that might have accounted for treatment failure in these patients is infection with another pathogenic agent, as it is known that polymicrobial co-infection associates with higher mortality in IA [53]. In both cases, we did not detect any evidence to suspect co-infection; thus, although we cannot completely rule it out, co-infection does not seem to be a factor in the therapeutic failure suffered by these patients.

In summary, we propose that azole persistence in *A. fumigatus* isolates may have contributed, along with other factors, to treatment failure in the presented cases. To our knowledge, this is the first report presenting a potential implication of antifungal persistence in clinical therapeutic failure. While mortality rates are always very high in these patient groups, it is known that the presence of resistant isolates is associated with an excess overall mortality in patients with IA [39]. Our results support the hypothesis that the presence of persister isolates may similarly increase mortality, due to a lower efficacy of treatment. We acknowledge that this study is limited and, consequently, no formal conclusions can be made at this point; however, we would like to advocate for the potential relevance of antifungal persistence in the treatment of *A. fumigatus* infections and exhort the community to consider and study this phenomenon and its consequences.

## Figures and Tables

**Figure 1 jof-09-00805-f001:**
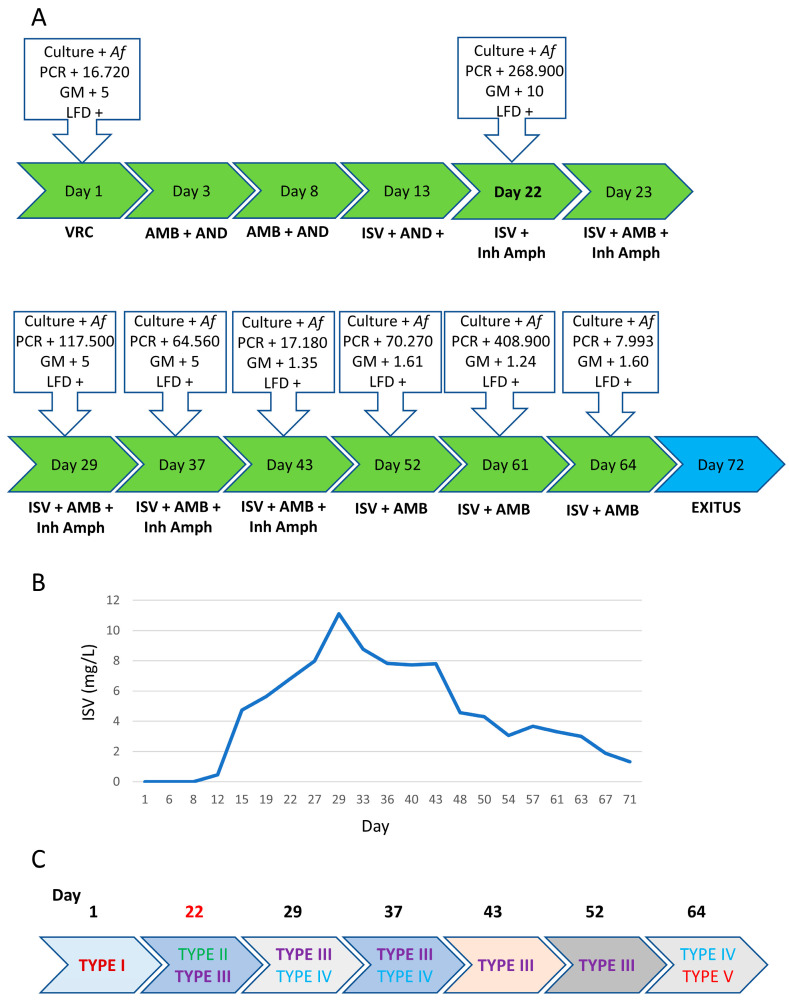
Progression of infection in patient 1. (**A**) Timeline showing the antifungal therapy administered to the patient and the results of the sample analyses performed throughout the course of infection. (**B**) Retrospective monitoring of the isavuconazole blood levels during the period of therapy. (**C**) Schematic overview of the time of detection for each of the five different *A. fumigatus* genotypes detected in patient 1.

**Figure 2 jof-09-00805-f002:**
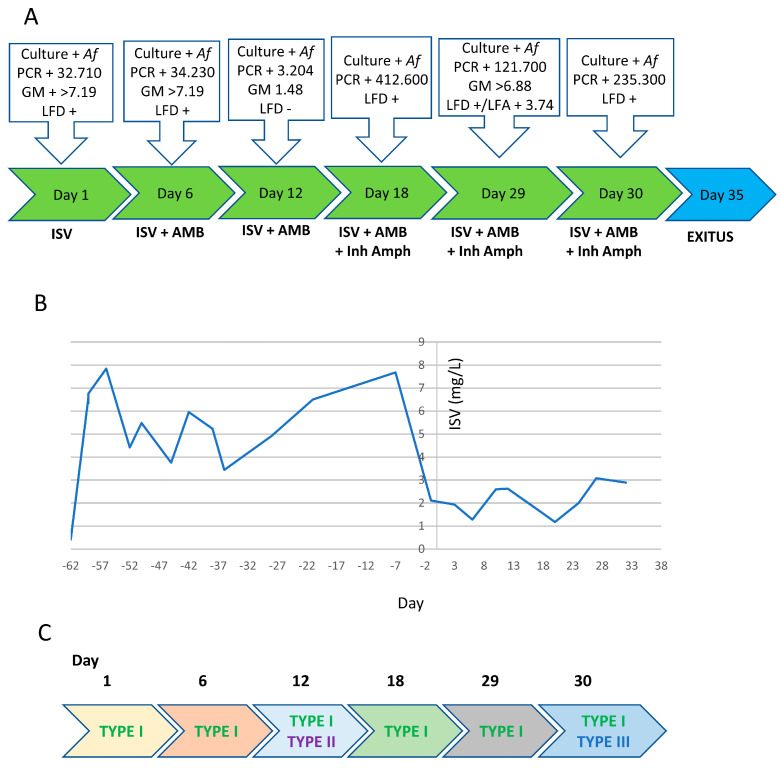
Progression of infection in patient 2. (**A**) Timeline showing the antifungal therapy administered to the patient and the results of the sample analyses performed throughout the course of infection. (**B**) Retrospective monitoring of the isavuconazole blood levels during the period of therapy. (**C**) Schematic overview of the time of detection for each of the three different *A. fumigatus* genotypes detected in patient 2.

**Table 1 jof-09-00805-t001:** Characteristics of *A. fumigatus* isolates from patient 1. * SP = sputum, TA = tracheal aspirate, BAL = bronchoalveolar lavage.

Isolate	Sample Type *	Day	MIC (mg/L)					Genotype					Persistence VRC	Persistence ISV	Resistance Mechanism	
								TRESPERG								
			AmB	ITC	VRC	POS	ISV	CSP	MP2	CFEM	ERG4B				*cyp51A*	*cyp51B*
**H-2067**	**BAL**	**1**	0.5	0.5	0.5	0.06	1	**t02**	**m1.1**	**c09**	**e05**	**TYPE I**	-	-	WT	WT
**H-2099**	SP	22	0.5	0.125	0.5	0.06	0.5	**t04A**	**m5.1**	**c09**	**e06**	**TYPE II**	-	-	WT	WT
**H-2100**		22	1	0.5	1	0.125	1	**t04A**	**m5.1**	**c09**	**e06**	**TYPE II**	-	-	WT	WT
**H-2101**		22	1	0.5	0.5	0.06	1	**t03**	**m1.1**	**c07**	**e07**	**TYPE III**	+	+	WT	WT
**H-2116**	SP	29	0.5	0.5	0.5	0.03	1	**t03**	**m1.1**	**c07**	**e13**	**TYPE III**	**+**	+	WT	WT
**H-2117**		29	0.5	0.25	0.5	0.03	1	**t11**	**m1.8**	**c07**	**e13**	**TYPE IV**	+	+	WT	WT
**H-2133**	SP	37	1	0.5	0.5	0.06	2	**t11**	**m1.8**	**c07**	**e07**	**TYPE IV**	+	+	WT	WT
**H-2134**		37	0.5	0.125	0.5	0.03	0.5	**t03**	**m1.1**	**c07**	**e07**	**TYPE III**	+	+	WT	WT
**H-2154**	SP	43	1	0.5	1	0.06	1	**t03**	**m1.1**	**c07**	**e07**	**TYPE III**	+	+	WT	WT
**H-2155**		43	1	0.125	0.5	0.03	0.5	**t03**	**m1.1**	**c07**	**e07**	**TYPE III**	+	+	WT	WT
**H-2165**	TA	52	1	0.5	1	0.06	1	**t03**	**m1.1**	**c07**	**e07**	**TYPE III**	+	+	WT	WT
**H-2173**		52	1	0.125	0.5	0.03	0.5	**t03**	**m1.1**	**c07**	**e07**	**TYPE III**	+	+	WT	WT
**H-2226**	BAL	64	1	0.5	0.5	0.125	1	**t11**	**m1.8**	**c07**	**e13**	**TYPE IV**	+	-	WT	WT
**H-2285**		64	1	>8	8	0.5	8	**t10**	**m1.1**	**c07**	**e05**	**TYPE V**			TR_34_/L98H	WT

**Table 2 jof-09-00805-t002:** Characteristics of *A. fumigatus* isolates from patient 2. * SP = sputum, TA = tracheal aspirate.

Isolate	Sample Type *	Day	MIC (mg/L)					Genotype					Persistence VRZ	Persistence ISV	Resistance Mechanism	
								TRESPERG								
			AmB	ITC	VRC	POS	ISV	CSP	MP2	CFEM	ERG4B				*cyp51A*	*cyp51B*
**H-3161**	**SP**	**1**	0.5	0.5	0.5	0.125	0.5	**t04A**	**m3.4**	**c22b**	**e11**	**TYPE I**	-	-	WT	WT
**H-3168**	SP	6	0.5	0.5	0.5	0.125	0.5	**t04A**	**m3.4**	**c22b**	**e11**	**TYPE I**	+	+	WT	WT
**H-3178**		6	0.5	0.5	0.5	0.25	1	**t04A**	**m3.4**	**c22b**	**e11**	**TYPE I**	-	+	WT	WT
**H-3180**	SP	12	0.5	0.5	0.5	0.125	0.5	**t04A**	**m3.4**	**c22b**	**e11**	**TYPE I**	+	+	WT	WT
**H-3181**		12	0.5	0.5	0.25	0.125	0.25	**t02**	**m5.1**	**c08A**	**e11**	**TYPE II**	+	+	WT	WT
**H-3183**		12	0.5	0.5	0.25	0.125	0.5	**t04A**	**m3.4**	**c22b**	**e11**	**TYPE I**	-	+	WT	WT
**H-3184**		12	0.5	0.5	0.5	0.125	0.5	**t04A**	**m3.4**	**c22b**	**e11**	**TYPE I**	-	-	WT	WT
**H-3185**		12	0.5	0.5	0.5	0.125	0.5	**t04A**	**m3.4**	**c22b**	**e11**	**TYPE I**	-	-	WT	WT
**H-3192**	TA	18	0.5	0.5	0.5	0.125	0.5	**t04A**	**m3.4**	**c22b**	**e11**	**TYPE I**	-	+	WT	WT
**H-3200**	TA	29	0.5	0.5	0.25	0.125	0.5	**t04A**	**m3.4**	**c22b**	**e11**	**TYPE I**	-	+	WT	WT
**H-3201**	TA	30	0.5	0.5	0.25	0.125	0.5	**t02**	**m1.1**	**c09**	**e11**	**TYPE III**	+	+	WT	WT
**H-3202**		30	1	0.5	0.25	0.125	0.5	**t04A**	**m3.4**	**c22b**	**e11**	**TYPE I**	-	+	WT	WT

## Data Availability

All data (excluding confidential patient information) can be found in this article or in the Appendix A. Access to the isolates can be granted upon reasonable request to the corresponding authors.

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
