# Peer review of "Potential Implication of Azole Persistence in the Treatment Failure of Two Haematological Patients Infected with Aspergillus fumigatus"

_jof, 2023, doi:10.3390/jof9080805_

Round 1
Reviewer 1 Report
Peláez-García de la Rasilla and colleagues submit a manuscript regarding the potential implication of azole persistence in the treatment failure of two hematological patients infected with Aspergillus fumigatus. The figures and data analysis are very nice.
Comments:
- The currently accepted terminology for “hematopoietic stem cell transplant (allo-HSCT)” is “allogeneic hematopoietic cell transplant (allo-HCT).” Please update.
- Note spelling issue: “acyclovir” is spelled “acyclovir”.
- Note that “septrin forte” is really “trimethoprim-sulfamethoxazole.
- For patients 1 and 2, please list the dose and duration of POS (was it the liquid solution or the extended-release tablet, and at what amount?). For patient 1 in particular, was POS ever stopped before the diagnosis of fungal infection?
- In the introduction, reference 2 is outdated, since much has changed about aspergillosis since publication in 2006. Favor updating reference, such as to the N Engl J Med article in 2021 Oct 14;385(16):1496-1509. doi: 10.1056/NEJMra2027424.
- You are aware that treatment of steroid-refractory GVHD (i.e., needing to advance to ruxolitinib) is associated with more infections than simple steroid-sensitive GVHD (reference Transplant Cell Ther 2022 Aug;28(8):509.e1-509.e11. doi: 10.1016/j.jtct.2022.05.008.)? This could be referenced to the patient one’s predisposition to fungal infection.
- The word “Mycobacterium” should not have a capital letter or italics.
- What is TA? Is it a tracheal aspirate? Is that the same as a sputum suctioned from an endotracheal tube?
- What is TACAR?
- An abrupt drop in ISV blood levels could result from introducing a new drug to the patient’s regimen, which then changes the metabolism of ISV. Was anything new given to the patient?
- In the abstract, “despite been infected, received correct antifungal treatment and reached therapeutic levels of the azoles, failed treatment” has grammar inconsistency; suggest “despite having been infected, receiving the correct antifungal treatment, and reaching therapeutic levels of the azoles, failed treatment.” It would be good if a native English speaker could read through and assess grammar throughout the manuscript, as there are many minor errors.
Grammar needs minor but frequent improvement
Author Response
REVIEWER 1
Peláez-García de la Rasilla and colleagues submit a manuscript regarding the potential implication of azole persistence in the treatment failure of two hematological patients infected with Aspergillus fumigatus. The figures and data analysis are very nice.
We would like to thank this reviewer for the time taken to evaluate our work, and for the valuable comments and corrections proposed. We have addressed them all, as it is explained in detail below
Comments:
- The currently accepted terminology for “hematopoietic stem cell transplant (allo-HSCT)” is “allogeneic hematopoietic cell transplant (allo-HCT).” Please update.
Thank you for this correction, it has been updated (lines 154, 163, 166, 262 and 271)
- Note spelling issue: “aciclovir” is spelled “acyclovir”.
Thank you for noticing this typo, it has been corrected (line 160)
- Note that “septrin forte” is really “trimethoprim-sulfamethoxazole.
We thank the reviewer for drawing attention to this mistake, it has been modified (lines 160-161)
- For patients 1 and 2, please list the dose and duration of POS (was it the liquid solution or the extended-release tablet, and at what amount?). For patient 1 in particular, was POS ever stopped before the diagnosis of fungal infection?
According to the reviewer’s suggestion, we have indicated in the text the format and dose of the POS prophylaxis (lines 160 and 266). The first patient was under POS prophylaxis just until the extensive immunosuppressive regimen administered to treat aGVHD stopped. We have included this information in line 160. Patient 2 was under POS prophylaxis until it was substituted for ISV, as detailed in the text (line 273).
- In the introduction, reference 2 is outdated, since much has changed about aspergillosis since publication in 2006. Favor updating reference, such as to the N Engl J Med article in 2021 Oct 14;385(16):1496-1509. doi: 10.1056/NEJMra2027424.
We are deeply grateful to the reviewer for providing a more recent reference. We have deleted the outdated reference and substituted it for the suggested one (lines 475-476)
- You are aware that treatment of steroid-refractory GVHD (i.e., needing to advance to ruxolitinib) is associated with more infections than simple steroid-sensitive GVHD (reference Transplant Cell Ther 2022 Aug;28(8):509.e1-509.e11. doi: 10.1016/j.jtct.2022.05.008.)? This could be referenced to the patient one’s predisposition to fungal infection.
The reviewer is right that ruxolitinib may favour fungal infections.
In addition to the reference kindly highlighted by the reviewer, Zeiser et al (1) also reported the increased risk of fungal infection in patients receiving ruxolitinib compared with the control regimen (11.5% vs 5.7%), indicating that antifungal prophylaxis needs to be administered for patients with acute or chronic GVHD treated by ruxolitinib.
In agreement with the literature, the patient was under POS prophylaxis during this treatment to cover for the additional risk of aspergillosis
As suggested by the reviewer, we have included a sentence to stress the effect of ruxolitinib on the risk of infection (line 159) and added the two references.
- Robert Zeiser, M.D., et al., Ruxolitinib for Glucocorticoid-Refractory Chronic Graft-versus-Host Disease N Engl J Med 2021; 385:228-238. DOI: 10.1056/NEJMoa2033122
- The word “Mycobacterium” should not have a capital letter or italics.
We thank the reviewer for this correction. We have edited “Mycobacteria” to write it correctly as “mycobacteria” (line 276)
- What is TA? Is it a tracheal aspirate? Is that the same as a sputum suctioned from an endotracheal tube?
TA is a tracheobronchial aspirate. The collection of TA by the traditional technique was performed according to standard procedure, that is, using a 12 French (Fr) siliconized polyvinyl chloride (PVC) tracheal aspiration probe. This was introduced through the ETT until resistance was encountered (level of the carina in the trachea), and retracted approximately 2cm. This was followed by the release of the vacuum and the probe was delicately removed using turning movements, from which the secretion was aspirated into a sterile polypropylene collector tube. No saline solution was used for either sample to liquefy secretions, and strictly aseptic principles were followed.
- What is TACAR?
We apologise for this mistake, TACAR is the Spanish abbreviation for High-resolution computed tomography (HRCT). We have corrected this in the text (lines 282 y 295)
- An abrupt drop in ISV blood levels could result from introducing a new drug to the patient’s regimen, which then changes the metabolism of ISV. Was anything new given to the patient?
The reviewer is right that some drugs can cause a change in the metabolism of ISV, explaining the abrupt drop observed in patient 2. However, the only drug introduced at the point of inflection was amphotericin-B, which is not expected to affect ISV levels.
Please, see the response to reviewer 2 to find more details about the decrease in ISV levels in both patients.
- In the abstract, “despite been infected, received correct antifungal treatment and reached therapeutic levels of the azoles, failed treatment” has grammar inconsistency; suggest “despite having been infected, receiving the correct antifungal treatment, and reaching therapeutic levels of the azoles, failed treatment.” It would be good if a native English speaker could read through and assess grammar throughout the manuscript, as there are many minor errors.
We thank the reviewer for the suggestion, we have edited the text accordingly.
Following the reviewer’s advice, we have asked a native English speaker colleague to assess the language, which has been extensively edited throughout the manuscript.
Reviewer 2 Report
The work is very interesting and allows us to consider a new hypothesis for treatment failures.
It illustrates this hypothesis with two clinical cases.
The methodology used to highlight a persistent strain has already been published.
For both patients, there is a significant isavuconazole levels decrease during treatment. Is there a clinical explanation?
Author Response
REVIEWER 2
The work is very interesting and allows us to consider a new hypothesis for treatment failures.
It illustrates this hypothesis with two clinical cases.
We would like to thank the reviewer for the positive evaluation of our work
The methodology used to highlight a persistent strain has already been published.
As far as we know, this exact protocol, which is optimised for the detection of persistence in filamentous fungi, has not been published. If the reviewer referred to our previous publication (https://doi.org/10.1128/spectrum.04770-22), it is relevant to say that the methodology described in this article differs significantly from those used in that study.
For both patients, there is a significant isavuconazole levels decrease during treatment. Is there a clinical explanation?
This is a thoughtful and appropriate comment. Regrettably, in these cases we do not have direct evidence to explain the decreases in ISV levels and can only speculate potential reasons.
It is expectable that deterioration of the patients’ conditions reduces the absorption and affect the drug levels, which could explain the drops. Indeed, Melchio et al presented an abstract at the ECCMID 2023 Congress, demonstrating that ICU patients had significantly lower levels of ISV compared to non-ICU population, showing that TDM of ISV for efficacy should be considered in ICU (1). We have included this reference in the discussion (lines 408-409).
Additional studies have shown other variabilities to consider. For instance, pharmacokinetics (PK) modelling showed that renal replacement therapy (RRT) is significantly associated with under exposure, partially explaining clearance variability. In this study, the Monte Carlo simulations suggested that the recommended dosing regimen of ISV did not achieve the trough target of 2 mg/L in a timely manner (72 h) (2). In another study, it was found that the standard dose of ISV did not achieve therapeutic concentration in a patient receiving concomitant ECMO support (3). Finally, inflammation of the gastrointestinal tract caused by GVHD may have contributed to the decrease in absorption.
- Melchio M., Mikulska M., Nadir U., Giacobbe D.R., Magnasco L., Limongelli A., Sepulcri C., Dentone C., Portunato F., Balletto E., Vena A., Miletich F. and Bassetti M. Lower blood levels of isavuconazole in critically ill patients compared to other populations: possible need for TDM. Abstract O0217. ECCMID Copenhagen 15-18 April 2023.
- Perez L, Corne P, Pasquier G, Konecki C, Sadek M, Le Bihan C, Klouche K, Mathieu O, Reynes J, Cazaubon Y. Population Pharmacokinetics of Isavuconazole in Critical Care Patients with COVID-19-Associated Pulmonary Aspergillosis and Monte Carlo Simulations of High Off-Label Doses. J Fungi (Basel). 2023 Feb 6;9(2):211. doi: 10.3390/jof9020211.
- Miller M., Kludjian G., Mohrien K and Morita K. Decreased isavuconazole trough concentrations in the treatment of invasive aspergillosis in an adult patient receiving extracorporeal membrane oxygenation support, American Journal of Health-System Pharmacy, Volume 79, Issue 15, 1 August 2022, Pages 1245–1249, https://doi.org/10.1093/ajhp/zxac043
Reviewer 3 Report
Dear Authors
I have read your work and I liked it very much.
It seems to me well developed and with very important results.
congratulations for your work.
thank you very much
Author Response
We are grateful to the reviewer for the time taken to assess our work and for the positive evaluation and comments.